# Research on Heat Transfer Performance of Micro-Channel Backplane Heat Pipe Air Conditioning System in Data Center

**Liping Zeng [1,*], Xing Liu [1], Quan Zhang [2], Jun Yi [2], Xianglong Liu [1] and Huan Su [1]**

1   Hunan Engineering Research Center of Energy Saving and material Technology of Green and Low Carbon Building, Hunan Institute of Engineering, Xiangtan 411104, China; edu_18163979051@163.com (X.L.); 31014@hnie.edu.cn (X.L.); 17029@hnie.edu.cn (H.S.)
2   College of Civil Engineering, Hunan University, Changsha, Hunan 410082, China; quanzhang@hnu.edu.cn (Q.Z.); hezhiwen619@163.com (J.Y.)
*   Correspondence: zengliping223@163.com; Tel.: +86-180-0842-6267

**Abstract:** This paper deals with the heat transfer performance of a micro-channel backplane heat pipe air conditioning system. The optimal range of the filling rate of a micro-channel backplane heat pipe air conditioning system was determined in the range of 65–75%, almost free from the interference of working conditions. Then, the influence of temperature and air volume flow rate on the heat exchange system were studied. The system maximum heat exchange is 7000–8000 W, and the temperature difference between the inlet and outlet of the evaporator and the condenser is almost 0 °C. Under the optimum refrigerant filling rate, the heat transfer of the micro-channel heat pipe backplane system is approximately linear with the temperature difference between the inlet air temperature of the evaporator and the cooling distribution unit (CDU) inlet water temperature in the range of 18–28 °C. The last part compares the heat transfer characteristics of two refrigerants at different filling rates. The heat transfer, pressure, and refrigerant temperature of R134a and R22 are the same with the change of filling rate, but the heat transfer of R134a is lower than that of R22. The results are of great significance for the operational control and practical application of a backplane heat pipe system.

**Keywords:** micro-channel; backplane heat pipe; optimal filling rate; heat transfer performance

## 1. Introduction

Given the rise of artificial intelligence (AI), cloud computing, communication, big data, Internet, finance, e-commerce, e-government, and other data-driven methods, the role of data centers will become more prominent. Therefore, finding more efficient ways to operate them is essential [1]. From 2013 to 2018, the scale of China's data center has grown nearly three times, with an average annual growth rate of more than 30%. By 2019, the scale of China's data center market will reach US $21.81 billion, accounting for 30.1% of the global total scale. Compared with the global market, the size of the data center market increased from US $28.44 billion in 2013 to US $62.6 billion in 2018, with a compound growth rate of only 17.1% [2]. As data center equipment is intensive and energy consumption is high, data room designers and equipment manufacturers are actively seeking various technical means to reduce power consumption and improve energy efficiency. As a new type of separated heat pipe, the micro-channel backplane heat pipe system is a kind of efficient heat exchange element which uses the temperature difference to drive the phase change heat exchange of refrigerant. In a certain environment, it does not need a compressor. It has the advantages of simple structure, flexible layout, strong adaptability, good sealing, and multi-fluid heat transfer ability. At the same time, it has a large surface area to volume ratio, a large heat exchange area effectively utilized, and high heat

transfer efficiency [3–5]. Therefore, the study of the micro-channel structure has become a hot topic in recent years.

Ling [6,7] carried out experiments on the micro-channel separated heat pipe system for a base station, and the results showed that under different indoor conditions, the cooling capacity and energy efficiency ratio (EER) of the system increased with the increase of indoor and outdoor temperature difference. Using a water-cooled multi-connected heat pipe system, the cooling capacity of the system can reach 6100–6200 W [8]. Wu et al. [9,10] studied the micro-channel separated heat pipe under standard conditions and different outdoor temperature conditions; the experimental results showed that the internal pressure of the evaporator first decreased and then increased with the increase of the refrigerant filling rate, and the superheat and supercooling at the inlet and outlet of the evaporator were close to 0 °C within the range of the optimal refrigerant filling rate. Jin [11,12] studied the micro-channel separated heat pipe with R134a as the medium, and the experimental results showed that the temperature difference increased from 10 °C to 20 °C, the heat exchange rate increased by 106%, the height difference increased from 0.75 m to 1.25 m, the increase rate was 267%, the system coefficient of performance (COP) was 4.66–13.9, and the power usage effectiveness (PUE) could reach 1.71. The power saving rate was 44.7%. Although scholars have done a lot of research on the micro-channel heat pipe, there is little research on the micro-channel backplane heat pipe system. Chen [13] analyzed the heat transfer characteristics of the backplane heat pipe air conditioning system, and the research showed that backplane air conditioning can automatically adjust the heat exchange according to the environmental changes. Sun [14–16] et al. concluded that the backplane heat pipe system adapts to the changes of different conditions, and the server should be placed in the middle and lower part as much as possible in order to obtain the maximum heat exchange capacity of the back plate. Compared with traditional precision air conditioning, the anti-condensation backplane heat pipe system can save 20–40% energy. Liu [17] analyzed the safety problems of the backplane heat pipe in the data center. It was found that when one fan in the system was damaged, it had little impact on the heat exchange of the system, and the temperature rise of the cabinet was 0.6–0.7 °C. Liu et al. [18,19] analyzed the unbalanced operation of the heat pipe backplane, and founded that the main reason for the imbalance of the backplane air conditioning system was the uneven load between cabinets. More backplane air conditioners do not undertake refrigeration, and those that undertake refrigeration experience severe dry burning of the upper part of the evaporator side, but this has little impact on the air inlet temperature of the overall cabinet. Luo et al. [20] studied the extreme working conditions of the heat pipe backplane in the data room and found that the temperature in the room was uniform during normal operation. When a heat pipe backplane fails, the air volume flow rate can be increased to keep the air inlet temperature stable, but increasing the air volume flow rate will directly cause the air outlet temperature of the backplane to rise, which has corresponding limitations. Ding et al. [21] measured and analyzed the application of heat pipe backplane in data center, and the results showed that the heat pipe backplane can use natural cold sources in transition season and winter, and the PUE in summer, winter, and transition season was 1.58, 1.20, and 1.38, respectively. To sum up, scholars have performed minimal research on the micro-channel heat exchanger as the condenser and evaporator of the separated heat pipe air conditioning system; most of the research is limited to the engineering application design experiment of the system in the data center.

This paper mainly studies the heat transfer performance of the micro-channel backplane heat pipe air conditioning system and conducts heat transfer performance experiments in a standard enthalpy difference laboratory. Through experiments, the micro-channel backplane heat pipe system is summarized and analyzed from the use effect. Parameters such as refrigerant pressure and temperature during system operation were measured, and the heat transfer effect of the system was analyzed from the perspective of the internal heat transfer mechanism. The refrigerant filling rate of the micro-channel backplane heat pipe air conditioning system was analyzed and the optimal liquid filling rate of the system under standard working conditions was measured. In addition, the environmental conditions may change during actual operation. In order to ensure that the system can always be in the best

operation state, the influence of different working conditions on the best refrigerant filling rate of the backplane heat pipe system was analyzed through experiments. The effects of different refrigerants on the heat transfer of the system under standard conditions and the heat transfer performance of the micro-channel backplane heat pipe system were also analyzed.

## 2. Experimental Method

### 2.1. Description of the Micro-Channel Backplane Heat Pipe Air Conditioning System

Backplane heat pipe air conditioning system as shown in Figure 1. The evaporator form of a backplane heat pipe system is the micro-channel heat pipe evaporator, and the condenser form is a copper brazed plate condenser cooling distribution unit (CDU). The chilled water is provided by an outdoor water chiller and cooling tower. The complete heat pipe backplane system consists of a back plate evaporator, gas pipe branch, CDU condensation section, liquid pipe branch, and other components. When the system is running, the refrigerant in the evaporation section of the back plate absorbs heat and becomes gaseous. The gaseous refrigerant rises through the gas pipe branch and enters the CDU condensation section. After heat exchange with the cooling water in the CDU, the gaseous refrigerant condenses into liquid. Under the effect of gravity, the refrigerant flows down through the liquid pipe branch and returns to the evaporation section of the back plate. Such repeated circulation operation can realize heat transfer.

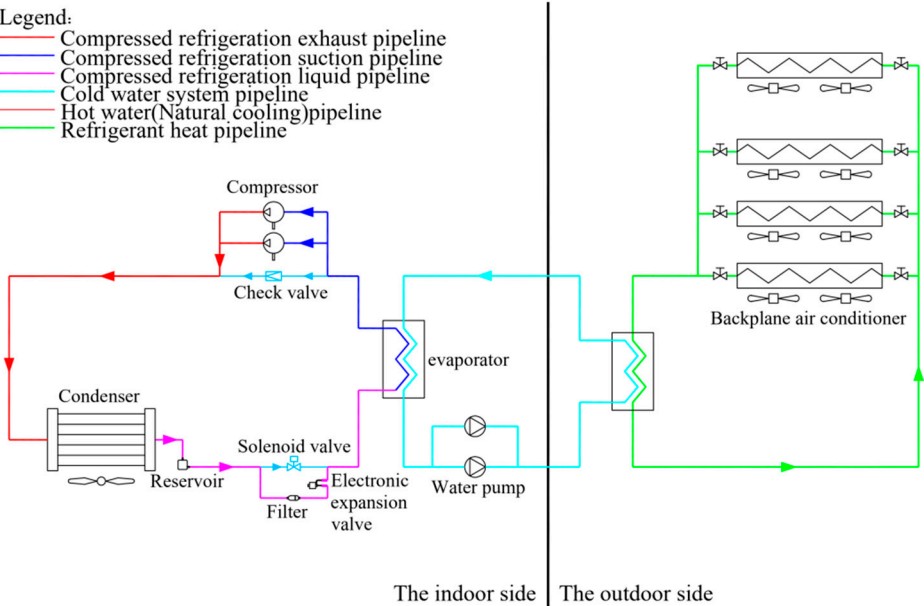

**Figure 1.** Backplane heat pipe air conditioning system.

### 2.2. Experimental Apparatus

The test unit includes a back plate evaporator side (with cabinet), CDU condensation side, riser, and downcomer. The experimental equipment included an enthalpy difference laboratory, which can create the indoor side temperature and humidity environment conditions required by the backplane heat pipe experiment, open and control the indoor water side equipment system, and provide the cold water supply temperature and flow required by the CDU cooling section experiment; a schematic diagram is shown in Figure 2 below. PT100 temperature sensor (Beijing Huakong Xingye Technology Development Co., Ltd., Beijing, China) was used to measure the refrigerant temperature at the inlet and outlet of the backplane heat pipe evaporation section and CDU condensation section. A K-type thermocouple temperature sensor (Changzhou Anbai Precision Instrument Co., Ltd., Changzhou, Jiangsu, China) was used to measure the wall temperature of the evaporation section of the backplane

heat pipe. The pressure sensor was used to measure the refrigerant pressure at the inlet and outlet of the evaporation section and the condensation section of the backplane heat pipe. Coriolis flowmeter (Jiangsu Kingstar Instrument Technology Co., Ltd., Huaian, Jiangsu, China) was used to measure the mass flow of refrigerant in the system. The experimental data measured by the experimental instrument was collected by Agilent data acquisition instrument (Beijing Dongfang Zhongke Integrated Technology Co., Ltd., Beijing, China) and connected to a computer for storage.

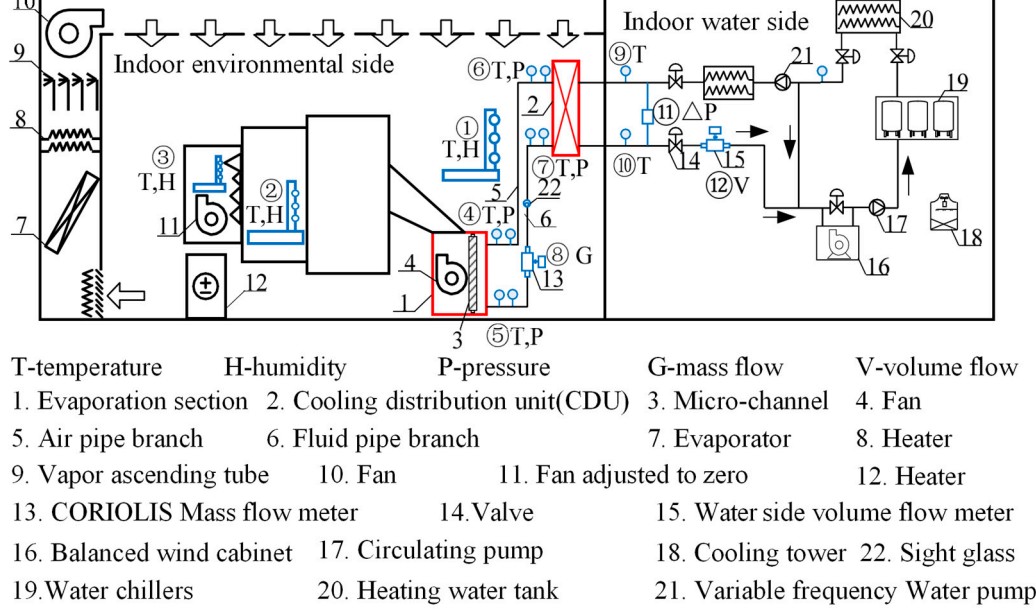

T-temperature      H-humidity      P-pressure      G-mass flow      V-volume flow
1. Evaporation section   2. Cooling distribution unit(CDU)   3. Micro-channel   4. Fan
5. Air pipe branch      6. Fluid pipe branch      7. Evaporator      8. Heater
9. Vapor ascending tube    10. Fan      11. Fan adjusted to zero      12. Heater
13. CORIOLIS Mass flow meter      14. Valve      15. Water side volume flow meter
16. Balanced wind cabinet    17. Circulating pump      18. Cooling tower   22. Sight glass
19. Water chillers      20. Heating water tank      21. Variable frequency Water pump

**Figure 2.** Schematic diagram of backplane heat pipe test platform of enthalpy difference laboratory.

### 2.3. Experimental Procedures

The standard operating conditions set in this experiment were as follows: indoor dry/wet bulb temperature was 35 °C/24 °C; the backplane circulating air volume was 1800 $m^3 \cdot h^{-1}$; and cooling water. The supply/return water temperature was 12 °C/17 °C, and the flow rate was 1.71 $m^3 \cdot h^{-1}$. The experimental content was carried out according to the following three parts. The first part is to analyze the effect of different filling rates on the heat transfer performance of the system and determine the optimal filling rate under different conditions. The second part is the heat transfer performance analysis of the system after determining the optimal filling rate. The third part is the comparison of the heat transfer performance of different refrigerants. The effects of different temperature, air volume flow rate, and other working conditions on the heat exchange performance of the system were analyzed. The other experimental conditions are set in Table 1. The experimental steps are the same as those of standard working conditions. The influence of R22 and R134a refrigerant on the heat exchange of the system was analyzed.

**Table 1.** Other experimental conditions.

| Parameters | Numerical Value |
|---|---|
| Refrigerant charge (kg) | 0.6, 0.8, 1.0, 1.2, 1.4, 1.6, 1.8, 2.0 |
| Indoor temperature (dry/wet bulb temperature) (°C) | 28/18.9, 30/20.9, 35/24.9, 40/29.9 |
| Air volume flow rate ($m^3 \cdot h^{-1}$) | 600, 800, 1000, 1200, 1400, 1600, 1800, 2000 |
| Chilled water inlet temperature (°C) | 10, 12, 14 |
| Chilled water flow ($m^3 \cdot h^{-1}$) | 1.71 |
| Working substance | R22, R134a |

The standard working conditions experimental steps are as follows:

(1) Through the enthalpy difference experiment control platform, the indoor environment temperature and humidity, water side water supply temperature and flow are set as the standard working conditions;

(2) Connect the measuring instrument, turn on the data acquisition instrument and the computer;

(3) Add refrigerant to the filling port, fill 0.6 kg for the first time, open the enthalpy difference test bench, set the data recording interval of each group as 20 s, and record the time for 30 min;

Repeat step (3) to complete the experiment and data recording of other charging conditions in turn, except for the standard conditions; other experimental conditions are shown in Table 1 below.

### 2.4. Experimental Uncertainty

During the test period, the cooling capacity of the evaporator section was calculated using the enthalpy difference between the indoor air and indoor air. Based on the uncertainty calculation method proposed by Mofft [22] and the major sensors' error in Table 2, the measurement uncertainty of evaporator section cooling capacity was 0.75%.

**Table 2.** Major sensors and their errors.

| Types | PT100 Temperature Sensor | K-Type Thermocouple Temperature Sensor | AKS32-060G2037 Pressure Sensor | DMF-1-3B Mass Flow Meter |
|---|---|---|---|---|
| | Temperature (°C) | Temperature (°C) | Pressure (MPa) | Flow Rate (kg·h$^{-1}$) |
| Errors | 0.1 | 0.1 | 0.0075 | 1 |
| Ranges | −30 to 200 | −200 to 260 | −0.1 to 2.4 | 0 to 300 |

## 3. Results and Discussion

### 3.1. Determination of the Optimal Filling Rate

Refrigerant filling rate is one of the most critical factors affecting the heat exchange performance of the backplane heat pipe air conditioning system. If the filling rate of the refrigerant is too small, the upper part of the evaporation section will be burnt dry, which will greatly reduce the heat exchange performance. If the filling rate of the refrigerant is too large, it will not only cause refrigerant waste, but also accumulate more liquid in the condensation section, which will reduce the heat exchange of the condensation section. Therefore, it is very important to determine an optimal range of refrigerant filling rate for the actual application of backplane heat pipe air conditioning.

Under the standard working condition, the heat exchange of the backplane air conditioning system changes with the refrigerant filling rate as shown in Figure 3 below. When the air conditioning system was charged with 0.6 kg refrigerant, the refrigerant filling rate was 27.9% and the heat exchange was very low, less than 1 kW. Since the system is in a lower refrigerant filling rate, there is only a small amount of liquid refrigerant at the bottom of the evaporator, and it will soon evaporate into a gaseous refrigerant, resulting in steam in the upper part of the evaporator with small heat capacity and without a liquid film covering the upper surface; so only through the steam heat exchange does the phenomenon of drying up occur, resulting in low heat exchange. With the refrigerant charge increases from 27.9% to 65.3% and the heat exchange of the system also significantly increases, reaching 7.8 kW, which is eight times the previous one. Due to this range of filling rate, with the increase of filling rate, the coverage of liquid film in the evaporator increases, so the heat exchange rate of phase change latent heat increases rapidly. However, as the refrigerant filling rate of the system is continuously increased, the heat exchange of the system is gradually reduced, because there is too much refrigerant in the evaporator, the length of the lower liquid pool is longer, and the length of the latent heat exchange is reduced. At the same time, the refrigerant at the inlet of the evaporator is supercooled, which will also

reduce the heat exchange of the evaporator. When the refrigerant filling rate of the system is higher than 80%, all the refrigerants in the evaporator are in a liquid phase state and cannot operate normally. Therefore, the range of 65–79% is the best range of the backplane heat pipe air conditioning system, and the heat transfer performance of the system is the best under the best range of the filling rate.

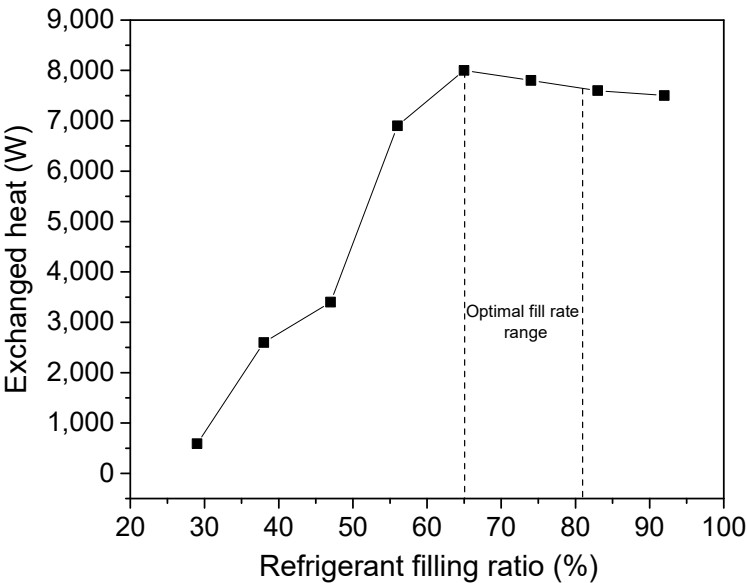

**Figure 3.** Change of heat exchange with refrigerant filling rate.

*3.2. Effect of Different Working Conditions on the Optimum Filling Rate*

3.2.1. The Influence of Different Temperature on the Best Filling Rate

Since the driving force of the operation of the backplane heat pipe air conditioning system depends on the temperature difference between the air in the evaporation section of the heat pipe and the outdoor condensation side, as well as the height difference between the evaporation section and the condensation section, the temperature condition in the machine room is generally maintained between 20 °C and 25 °C; but when the server in the room goes down, power is off or the outdoor weather conditions change dramatically, the temperature may change suddenly. Therefore, it is necessary to study the influence of different temperature conditions on the optimal refrigerant filling rate and heat exchange performance of the system, which is conducive to ensure that the backplane heat pipe air conditioning system is in the optimal operation state during the actual operation process, and the air outlet temperature meets the air supply temperature requirements of the data room. Table 3 shows the five different operating conditions set in the experiment. Under the five operating conditions, the experiments with different filling rates were carried out respectively.

**Table 3.** Different temperature conditions.

| Temperature °C | Condition 1 | Condition 2 | Condition 3 | Condition 4 | Condition 5 |
|---|---|---|---|---|---|
| Air inlet temperature of backplane | 35 | 30 | 40 | 35 | 35 |
| CDU inlet water temperature | 12 | 12 | 12 | 10 | 14 |

Figures 4 and 5 show the influence of the cooling capacity of the system filled with different refrigerant filling rates and the air outlet temperature at the evaporator side under different working conditions. It can be seen from the figures below that under different working conditions, the trend of the system changes but the refrigerant filling rate is the same, the range of the optimal refrigerant filling rate is about 65–79%, and the air outlet temperature of the evaporator backplane reaches the lowest, meeting the air supply requirements for the data room server. The experimental results show

that the inlet air temperature of the backplane and the inlet water temperature of the condenser have little effect on the range of the optimal refrigerant filling rate.

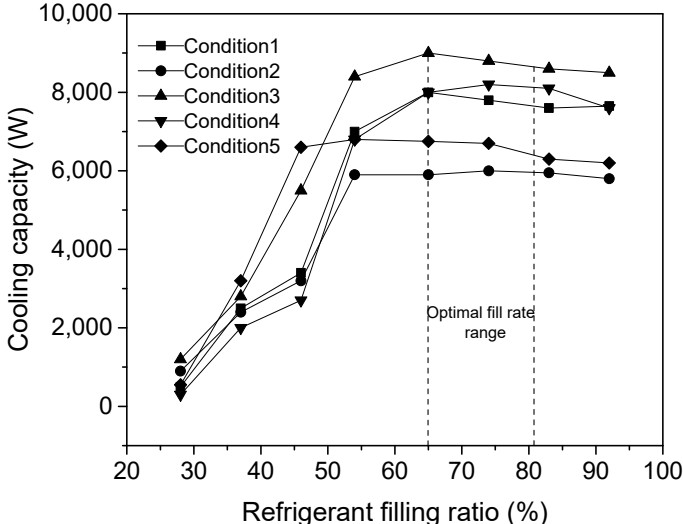

**Figure 4.** Relationship between cooling capacity and refrigerant filling rate under different temperature conditions.

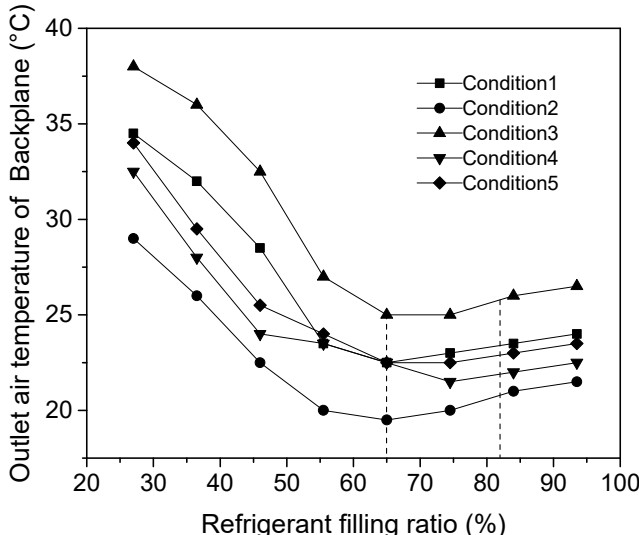

**Figure 5.** Relationship between liquid filling rate and backplane outlet temperature under different temperature conditions.

### 3.2.2. The Influence of Different Air Volume Flow Rate on the Optimal Filling Rate

The experiments of air volume flow rate (1800 $m^3 \cdot h^{-1}$) and air volume flow rate (1000 $m^3 \cdot h^{-1}$) under standard conditions are compared. Figure 6 shows the influence of liquid filling rate on cooling capacity under different air volume flow rates, and Figure 7 shows the influence of refrigerant filling rate on air outlet temperature of backplane under different air volume flow rates.

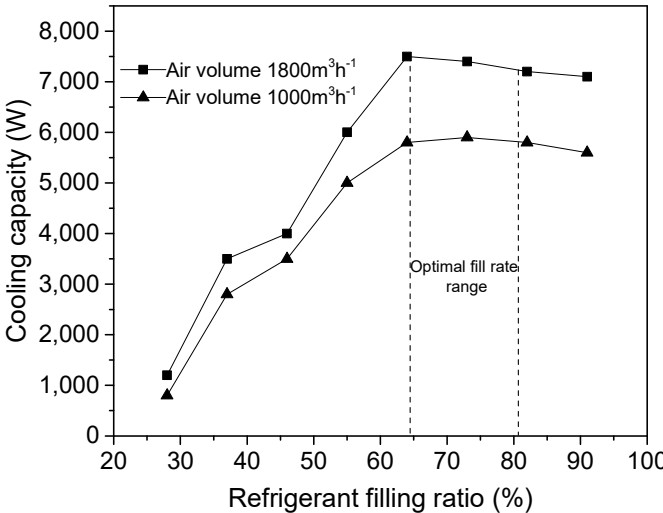

**Figure 6.** Relationship between cooling capacity and refrigerant filling rate under different air volume flow rate conditions.

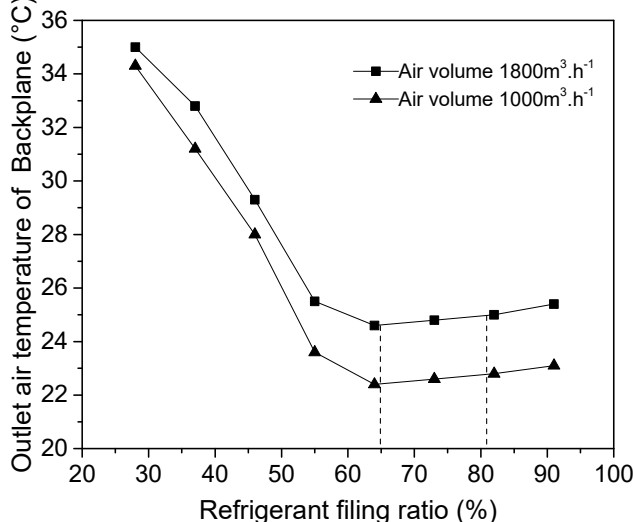

**Figure 7.** Relationship between refrigerant filling rate and backplane air temperature under different air volume flow rate conditions.

It can be seen from the Figures 6 and 7 that under two different air volume flow rate conditions, the change trend of heat exchange and air outlet temperature of the back plate is consistent with the increase of the liquid filling rate. When the refrigerant filling rate FR < 65%, the heat transfer increases rapidly with the increase of the refrigerant filling rate, and the air temperature of the evaporator side decreases rapidly. When the refrigerant filling rate FR = 65–79% (the optimal filling rate), the heat exchange and the air outlet temperature of the backplane are basically stable. At this time, the heat exchange of the whole system is the largest, the air outlet temperature of the backplane is the lowest, and the system is in the optimal working condition. When the refrigerant filling rate FR > 80%, the heat exchange begins to slow down, and the air temperature slowly increases. It shows that the change of air volume flow rate has little effect on the range of optimal filling rate of the backplane heat pipe system.

To sum up, when the external environment conditions change, the optimal filling rate range of the backplane heat pipe system will not change and will always remain at about 65–79%. This property of the backplane heat pipe is also of great significance for its practical engineering application. As long as the backplane heat pipe is filled to the optimal refrigerant filling rate range, the bottom plate heat

pipe is always in optimal operation regardless of external environmental conditions such as intake air temperature and inlet water temperature.

### 3.3. Effect of Different Refrigerants on Heat Transfer Characteristics

In order to study the influence of different refrigerant properties on the heat transfer of the system, R134a and R22 refrigerant are selected for comparison in this experiment, and the change of heat transfer with the refrigerant filling rate is shown in Figure 8 below. It can be seen from Figure 8 that the change of heat exchange rate of R134a refrigerant with refrigerant filling rate is basically the same as that of R22, and its optimal refrigerant filling rate range is about 65% to 79%, but the heat exchange rate of R134a is about 5% lower than that of R22 as a whole, because the latent heat of R22 gasification is slightly higher than that of R134a, and under the optimal refrigerant filling rate, the heat exchange of refrigerant is mainly two-phase heat exchange.

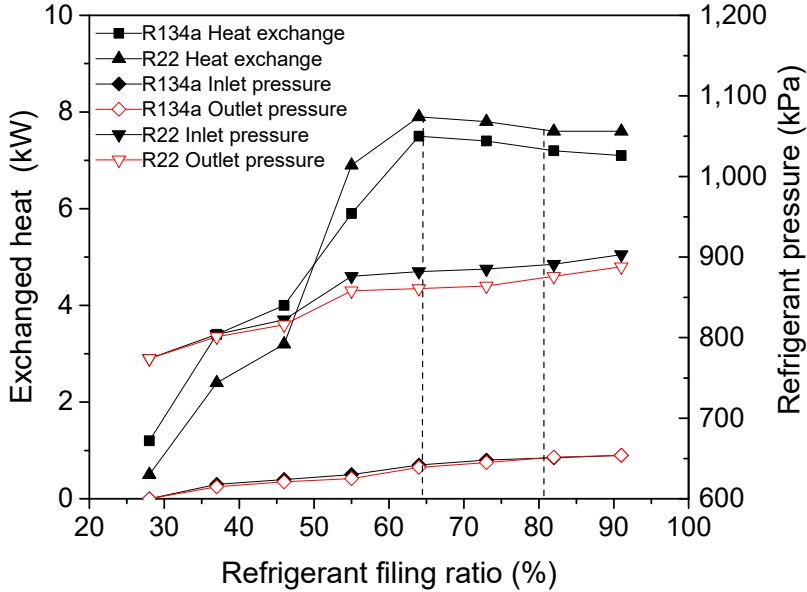

**Figure 8.** Heat exchange and refrigerant inlet and outlet pressure in evaporator.

The pressure at the inlet and outlet of R134a and R22 evaporators increase with the increase of refrigerant filling rate, and increase rapidly under the condition of a low refrigerant filling rate. Under the same conditions, the inlet and outlet pressure of R134a are lower than R22. At the optimal filling rate, R134a pressure is only about 580 kPa, while R22 pressure is about 870 kPa. The pressure drop of R134a and R22 in the evaporator is very small, the pressure drop is about 10–15 kPa. Figure 9 shows the temperature of R22 and R134a at the inlet and outlet of the evaporator at different refrigerant filling rates. It can be seen from the figures that the outlet temperature of the evaporator is close to 35 °C at low refrigerant filling rates, and the refrigerant at the outlet of the evaporator becomes superheated steam at low refrigerant filling rates. The heat exchange capacity is almost 0, and the refrigerant temperature is close to the inlet air temperature. With the increase of refrigerant filling rates, the temperature at the inlet and outlet of the refrigerant in the evaporator is close to the inlet air temperature which the temperature at the outlet is approaching gradually. When the refrigerant filling rate reaches 55%, then the temperature difference between the inlet and outlet of the two refrigerants approaches, and the heat exchange increases gradually. When the refrigerant filling rate reaches 65–79%, the refrigerant in the evaporator is in the state of a two-phase flow, at this time, the heat exchange is mainly latent heat, and the heat exchange is the largest. However, compared with R134a, the temperature difference between the inlet and outlet of R22 is larger. To sum up, in the optimal range of refrigerant filling (65–79%), the system has the largest heat exchange and the smallest temperature

difference between the inlet and outlet of the refrigerant in the evaporator. It can be seen that different refrigerants operate in different states in the backplane system of the heat pipe, and the influence of different refrigerants should be considered in the actual application of the backplane heat pipe system.

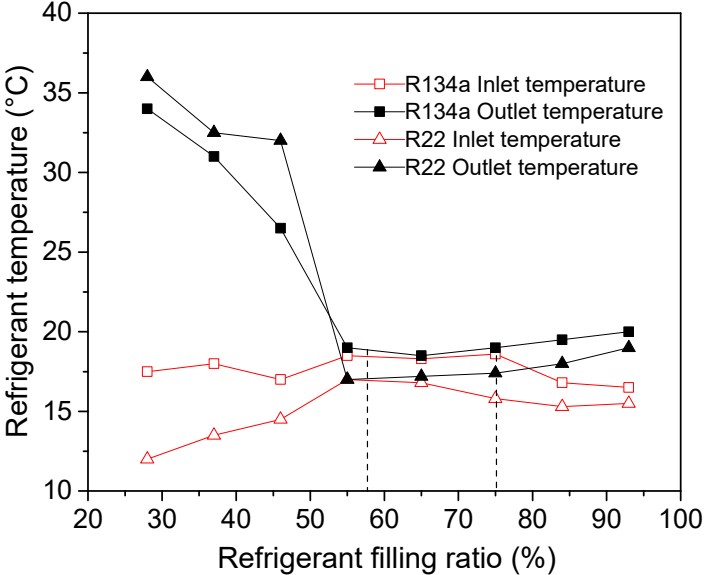

**Figure 9.** Temperature of refrigerant inlet and outlet in evaporator.

### 3.4. System Heat Exchange Performance Analysis

In the above experiments, the heat exchange changes with the refrigerant filling rate under five different air inlet temperatures, and water inlet temperature conditions are taken respectively. The heat exchange values at the optimal refrigerant filling rate are taken from the above results. Since the air volume flow rate (1800 m$^3$·h$^{-1}$) at the evaporator side and the cold water flow volume (1.71 m$^3$·h$^{-1}$) at the CDU side remain unchanged, conditions 1–5 can be described as the air inlet temperature and CDU of the evaporator The temperature difference of inlet water is 18–28 °C under five working conditions. Under the optimum filling rate, the change of heat exchange of the backplane heat pipe system with the temperature difference is shown in Figure 10 below.

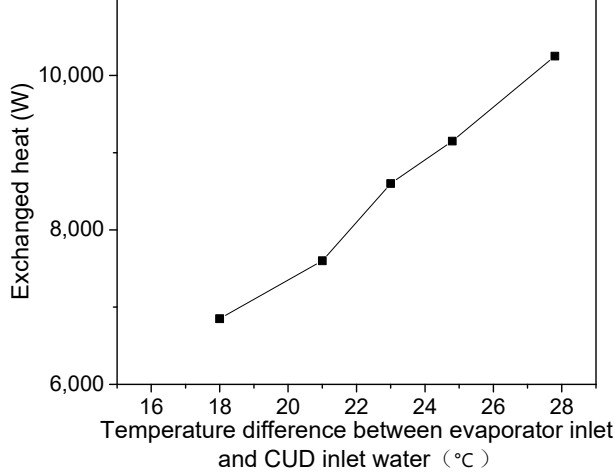

**Figure 10.** Change of heat exchange with temperature difference between inlet air of evaporator and inlet water of CDU.

It can be seen from Figure 10 that under the optimal refrigerant filling rate, when the air volume flow rate at the evaporator side and the water flow at the condenser side remain unchanged, the heat

exchange and the temperature difference between the air inlet temperature of the evaporator and the CDU water inlet temperature are almost linear.

For the analysis of the relationship between the heat exchange rate of heat pipe and the temperature difference between cold and hot ends, Zhu [23] used the $\varepsilon$-NTU method to calculate the heat transfer formula of the whole system under the optimal liquid filling rate of a separated heat pipe. The formula is as follows:

$$Q_e = Q_c = \frac{c_1 c_2}{c_1 + c_2} (t_{e,ain} - t_{c,win}) \tag{1}$$

$$c_1 = m_{ea} c_{p,a} \varepsilon_e, \ c_2 = m_{cw} c_{p,w} \varepsilon_c \tag{2}$$

$$\varepsilon_e = 1 - \exp(-NTU_e), \ \varepsilon_c = 1 - \exp(-NTU_c) \tag{3}$$

$$NTU_e = \frac{U_e A_e}{m_{ea} c_{p,a}}, \ NTU_c = \frac{U_c A_c}{m_{cw} c_{p,w}} \tag{4}$$

Li [24] et al. in a separated heat pipe experiment, and under the conditions of the optimal refrigerant filling rate and constant air volume flow rate at the evaporation and condensation side, adjusted the change of the inlet air temperature at the evaporator side within 5–42 °C and 20–55 °C at the condensation side. The experimental results show that the heat exchange is not related to the temperature change range but has a linear relationship with the temperature difference of the inlet air at the condenser side of the evaporator. From the above experimental results, under the condition of the optimal refrigerant filling rate, keeping the air volume flow rate at the evaporator side and the water volume at the condenser side unchanged, the heat exchange of the micro-channel backplane heat pipe system is approximately linear with the temperature difference between the air inlet temperature of the evaporator and the CDU water inlet temperature within a certain temperature range. The structure of backplane heat pipe is similar to that of separated heat pipe, but the condenser of backplane heat pipe system uses a CDU instead of micro-channel condenser. According to the analysis of the derivation process in Equation (1), the formula is also applicable to the backplane heat pipe system, and it is consistent with the experimental data results of the backplane heat pipe system.

### 3.5. $\varepsilon$-NTU Simplified Model

#### 3.5.1. Model Establishment

When the side air volume of the evaporator and the water flow of the condenser remain unchanged, the influence of the difference between the inlet air temperature of the evaporator and the inlet water temperature of the CDU on the heat transfer is analyzed by Equation (1). The formula is also applicable in the micro-channel backplane heat pipe system, but the condenser form of the backplane heat pipe system studied in this paper is a CDU plate heat exchanger. Combined with the experimental results, the heat transfer prediction model applicable to this system can be derived by using the $\varepsilon$-NTU method. The derivation process is as follows:

According to the experimental analysis above, when the system is in the best charging rate, import and export of refrigerant in the evaporator temperature generally approximates 0 °C and have good isothermal property, because the back of the evaporator and condenser heat pipe system are connected by an adiabatic section connection, So you can think, when the system is in the best charging rate, the refrigerant in the evaporator and condenser temperature are approximately equal, and evaporation temperature and condensation temperature degrees are nearly equal. So, evaporator side $m_r c_{p,r} \to \infty$, $C_{min} = m_{ea} c_{p,a}$, cold condenser side $m_r c_{p,r} \to \infty$, $C_{min} = m_{cw} c_{p,w}$.

First to calculate on the evaporator side:

$$\varepsilon_e = \frac{t_{e,ain} - t_{e,aout}}{t_{e,ain} - t_e} = 1 - \exp(-NTU_e) \tag{5}$$

$$NTU_e = \frac{U_e A_e}{m_{ea} c_{p,a}} \tag{6}$$

$$Q_e = m_{ea} c_{p,a} (t_{e,ain} - t_{e,aout}) \tag{7}$$

It can be obtained from Equation (5) above:

$$t_{e,aout} = -\varepsilon_e (t_{e,ain} - t_e) + t_{e,ain} \tag{8}$$

Substitute Equation (8) into Equation (7) to obtain:

$$Q_e = \varepsilon_e m_{ea} c_{p,a} (t_{e,ain} - t_e) \tag{9}$$

Calculate the condenser in the same way:

$$\varepsilon_c = \frac{t_{c,wout} - t_{c,win}}{t_c - t_{c,ain}} = 1 - \exp(-NTU_c) \tag{10}$$

$$NTU_c = \frac{U_c A_c}{m_{cw} c_{p,w}} \tag{11}$$

$$Q_c = m_{cw} c_{p,w} (t_{c,wout} - t_{c,win}) \tag{12}$$

It can be obtained from Equation (10) above:

$$t_{c,wout} = \varepsilon_c (t_c - t_{c,win}) + t_{c,win} \tag{13}$$

Substitute Equation (13) into Equation (12) to obtain:

$$Q_c = \varepsilon_c m_{cw} c_{p,w} (t_c - t_{c,win}) \tag{14}$$

The simultaneous Equations (9) and (14) can be obtained:

$$t_e = t_c = \frac{\varepsilon_a m_{ea} c_{p,a} t_{e,ain} + \varepsilon_c m_{cw} c_{p,w} t_{c,win}}{\varepsilon_a m_{ea} c_{p,a} + \varepsilon_c m_{cw} c_{p,w}} \tag{15}$$

By combining Equation (9) with Equations (14) and (15), the heat exchange prediction formula of the micro-channel backplane heat pipe system is obtained as follows:

$$Q = \frac{\varepsilon_a m_{ea} c_{p,a} \varepsilon_c m_{cw} c_{p,w} (t_{e,ain} - t_{c,win})}{\varepsilon_a m_{ea} c_{p,a} + \varepsilon_c m_{cw} c_{p,w}} \tag{16}$$

### 3.5.2. Model Validation

Based on the previous set of optimal filling rate conditions, the experimental data of different inlet air volume, inlet air temperature, and inlet water temperature were substituted into the model (Equation (16)) for calculation. The relative error between the calculated value and the experimental value is shown in Figure 11 below. As shown, the error of the prediction formula for the overall heat transfer of the system is about ±12%, which can basically meet the requirements for quickly predicting the heat transfer of the system in practical applications. When the backplane heat pipe is in the optimal refrigerant filling rate condition, it is considered that the system is in the optimal operation state, and its heat exchange can be calculated by the ε-NTU method. In actual operation, we only need to measure the inlet air temperature, inlet air flow, inlet water temperature, and inlet water flow, which are four physical quantities convenient for measurement. Then the corresponding heat transfer value of the system can be obtained by Equation (16). It is of great significance for real-time prediction and optimization control of heat transfer in a micro-channel backplane heat pipe system.

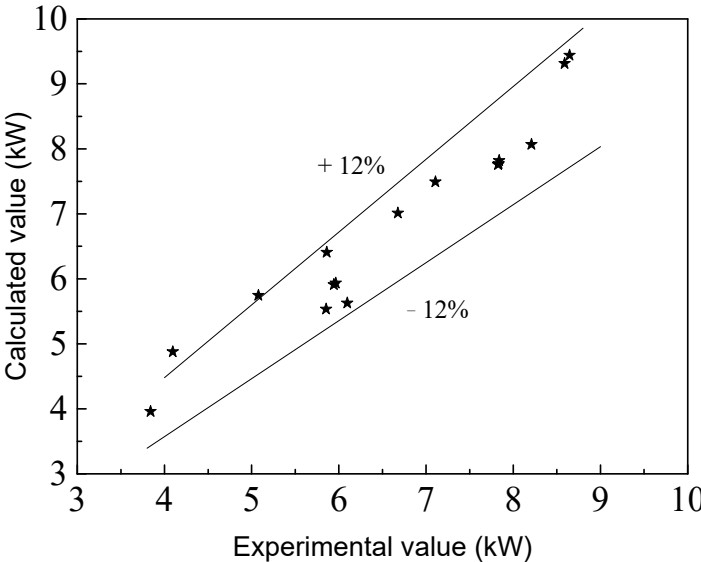

**Figure 11.** Relative error between calculated value and experimental value.

## 4. Conclusions

In this paper, a micro-channel backplane heat pipe system was proposed to reduce the cooling energy consumption of the data center. At the same time, the heat transfer characteristics of the micro-channel backplane heat pipe under different working conditions and different refrigerants are studied. The main conclusions are as follows:

(1) By comparing different working conditions and different refrigerants, the optimal filling rate of the system is basically in the range of 65–75%, almost free from the interference of working conditions. When the system is in the optimal filling rate, the heat exchange is the largest, reaching 7000–8000 W, the pressure drop is relatively small, 10–15 kPa, the air temperature is 22–24 °C, and its refrigeration capacity meets the load requirements of the data room.

(2) With the increase of refrigerant filling rate, the pressure and pressure drop at the inlet and outlet of the system will increase. The pressure drop in the evaporator is greater than that in the condenser, and the mass flow in the system will increase. At the best refrigerant filling rate, the temperature difference between the inlet and outlet of the evaporator and the condenser is almost 0 °C, and the refrigerant base in the evaporator is in a two-phase state.

(3) Under the optimal filling rate, when the medium flow rate at the evaporator side and condenser side is kept constant, the heat exchange of the micro-channel backplane heat pipe system is approximately linear with the temperature difference between the inlet air temperature of the evaporator and the CDU inlet water temperature within a certain temperature range.

(4) The heat exchange, pressure, and refrigerant temperature of R134a and R22 are the same with the change of filling rate, but under the same conditions, the heat exchange of R134a is lower than that of R22 on the whole. Under the optimum filling rate, the temperature at the inlet and outlet of the refrigerant in the evaporator is almost the same. The range of refrigerant filling rate corresponding to the maximum heat exchange and the minimum temperature difference between the inlet and outlet of the refrigerant in the evaporator is about 65–75%, which is the optimal refrigerant filling rate of the system.

(5) Using the $\varepsilon$-NTU method, a simplified prediction model suitable for the heat transfer of the system is derived. The error of the overall heat transfer prediction model of the system is about ±12%, which can basically meet the requirements for rapid prediction of the heat transfer of the system in practical applications. In actual operation, we only need to measure four physical quantities that can be easily measured to obtain the corresponding heat transfer value of the system.

**Author Contributions:** Research: L.Z., X.L. (Xing Liu), Q.Z., J.Y., X.L. (Xianglong Liu) and H.S.; Writing: L.Z., X.L. (Xing Liu), Q.Z., J.Y., X.L. (Xianglong Liu) and H.S. All authors have read and agreed to the published version of the manuscript.

**Funding:** The authors are grateful for supported by construct program of applied specialty disciplines in Hunan province (Hunan Institute of Engineering). The present study was supported by the National Key R&D program of China (2018YFE0111200) and (2016YFE0114300), Hunan provincial key projects (No.2016JJ5011), the provincial Natural Science Youth Foundation of Hunan, China (No.2018JJ3102), and Provincial Natural Science Foundation of Hunan, China (No. 2018JJ2081), Scientific Research Fund of Hunan Provincial Education Department, China (No.17B064), Project supported by Provincial Natural Science Foundation of Hunan, China (No. 2018JJ4040).

**Conflicts of Interest:** The authors declare no conflict of interest.

## Nomenclature

| | |
|---|---|
| $Q_e$ | Evaporators exchange heat (W) |
| $Q_c$ | Condenser heat transfer capacity (W) |
| $\varepsilon_e$ | Evaporator efficiency |
| $\varepsilon_c$ | Condenser efficiency |
| $t_{c,win}$ | Air inlet temperature at condenser side (°C) |
| $t_{c,wout}$ | Condenser side air outlet temperature (°C) |
| $t_{e,aout}$ | Evaporator side air outlet temperature (°C) |
| $A_e$ | Evaporator heat transfer area ($m^2$) |
| $A_c$ | Condenser heat transfer area ($m^2$) |
| $m_{ea}$ | Evaporator side air flow rate ($kg·s^{-1}$) |
| $m_{cw}$ | Air flow at condenser side ($kg·s^{-1}$) |
| $t_e$ | Evaporation temperature (°C) |
| $t_{e,ain}$ | Evaporator side air inlet temperature (°C) |
| $U_c$ | Condenser heat transfer coefficient ($W·(m^{-2}·K^{-1})$) |
| $U_e$ | Evaporator heat transfer coefficient ($W·(m^{-2}·K^{-1})$) |
| $c_{p,a}$ | Specific heat of the air inlet at the side of the evaporator ($J (kg·K^{-1})$) |
| EER | Cooling capacity (W)/input power (W). The higher the EER value, the more heat absorbed by evaporation in the air conditioner or the less power consumed by the compressor. |
| COP | Cooling capacity or heating capacity (W)/input power (W), the higher the COP value, the higher the system energy efficiency. |
| PUE | Total facility power/IT equipment power. The closer the PUE value is to 1.0, the higher the energy efficiency of the equipment room. |
| $\varepsilon$-NTU | Effective heat transfer unit number method. |

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
