# Peer review of "Research on Heat Transfer Performance of Micro-Channel Backplane Heat Pipe Air Conditioning System in Data Center"

_applsci, doi:10.3390/app10020583_

Round 1

Reviewer 1 Report

The paper deals with a very interesting subject. The literature review is quite complete and well explained. The methodology followed and the results are well organized and presented. English language is very good. Thus I recommend the paper for publication.

Author Response

Reviewer    #1

The paper deals with a very interesting subject. The literature review is quite complete and well explained. The methodology followed and the results are well organized and presented. English language is very good. Thus I recommend the paper for publication.

Response: Thanks the respectable reviewer for her/his valuable comment.

Reviewer 2 Report

The paper analyses the heat transfer performance of micro-channel backplane heat pipe air conditioning system. I do have a couple of comments mainly to increase the relevance of the manuscript.

Specific comments are presented below:

At page 2, Lines 42-43, it is specified: “In recent years, micro-channel structure has become a hot topic in the study of heat exchanger 42 performance”. What is the motivation of this aspect? Add the definite article before “micro-channel”. In this case, some language editing is needed in the manuscript. At page 4, the following sentences are unclearly presented: Lines 127-128 “after determining the optimal filling rate, the analysis of the heat transfer performance of the system”. The verb is missing. Lines 128-129 “the third part is the comparison of the heat transfer performance of different refrigerants”. Which are the first and the second parts? Lines 129-130 “Analyzed the influence of different temperature, air volume…”. Add the definite article “the” before “air volume”. Furthermore, the sentence begins with a verb, and the subject is missing. At page 5, Line 132 “The experimental steps are the same as those of the standard working conditions”. Which are the standard working conditions? At page 5, Lines 139-140 “Repeat step (3) to complete the experiment and data recording of other charging conditions in turn, except for the standard conditions, other experimental conditions are shown in Table 1 below”.

Where is Step 3 presented? Which is the difference between “the specific experimental conditions” (Page 4, Line 131) and “other experimental conditions”  (Page 5, Line 140)?  

In Table 1:

It is better to change “type” with “parameters”; The air volume is measured in m3/h? Replace “indoor temperature condition“ with “indoor temperature”. Page 6, Line 173, at the beginning of Section 3.2.1, replace “the influence” with “The influence”.   Page 7, Line 190, it is specified “It can be seen from the figure ….”. About what Figure is taking about? Page 8, Line 201, replace “air volume” with “air volume flow rate” considering that is talking about m3/h. Page 8, Line 203, replace “figure” with “Figure”. Page 8, Line 209, it is specified “It can be seen from the figure that under two different air volumes….”. Specify the number of the figure. Page 9, Line 228, replace “8” with “Figure 8”. The same remark at Line 269, pag.11. Page 10, Line 237, replace “;” with “.” at the end of the sentence. Furthermore, at Line 238 let a space between the sentences “than R22.At the optimal”. Page 11, Line 276, delete the point in “the optimal filling rate. Li [23] et al.”. Page 12, Line 293, is the “air inlet volume” or “air inlet volume flow rate”? Also, the sentence “need to be measured during the actual operation, which can” is incomplete. Furthermore, the sentence at the Lines 295-295 is too vague “The corresponding heat exchange of the system can be calculated quickly…”. Finally, at the Conclusions, the Authors should spread some words about the practical importance regarding the experimental results obtained in the literature. I strongly suggest that the Authors will carry out more studies to compare their results to that from other similar studies.

Author Response

Response to Reviewer 2 Comments

The paper analyses the heat transfer performance of micro-channel backplane heat pipe air conditioning system. I do have a couple of comments mainly to increase the relevance of the manuscript.

Response: Dear reviewer, thank you for your valuable suggestions. We applied your kind comments to the manuscript to improve the quality of it. Thank you for your time and consideration.

Point 1: At page 2, Lines 42-43, it is specified: “In recent years, micro-channel structure has become a hot topic in the study of heat exchanger 42 performance“. What is the motivation of this aspect?

Response 1: Thanks the respectable reviewer for your valuable comment. Lines 41-44, “It has many advantages. At the same time, it has a large surface area to volume ratio , a large heat exchange area for efficient use, and high heat transfer efficiency [3-5]. Therefore, the research on micro-channel structure has become a hot topic in recent years“.

Point 2: Add the definite article before “micro-channel”. In this case, some language editing is needed in the manuscript.

Response 2: Thanks the respectable reviewer for your valuable comment. The full text has been carefully checked and modified.

Point 3: Lines 128-129 “the third part is the comparison of the heat transfer performance of different refrigerants”. Which are the first and the second parts?

Response 3: Thank you for your valuable question. After revision, parts one and two have been listed in lines 133-136.

Point 4: Lines 129-130 “Analyzed the influence of different temperature, air volume…”. Add the definite article “the” before “air volume”. Furthermore, the sentence begins with a verb, and the subject is missing.

Response 4: Thanks the respectable reviewer for your valuable comment. The errors in the article have been corrected in lines 137-139.

Point 5: At page 5, Line 132 “The experimental steps are the same as those of the standard working conditions”. Which are the standard working conditions? At page 5, Lines 139-140 “Repeat step (3) to complete the experiment and data recording of other charging conditions in turn, except for the standard conditions, other experimental conditions are shown in Table 1 below”.Where is Step 3 presented? Which is the difference between “the specific experimental conditions” (Page 4, Line 131) and “other experimental conditions”  (Page 5, Line 140)?

Response 5: Thank you for your valuable question. Lines 130-132 “The standard working conditions set in this experiment are: indoor dry / wet bulb temperature is 35 ℃ / 24 ℃, the backplane circulating air volume is 1800 m3·h-1, and cooling water. The supply / return water temperature is 12 ℃ / 17 ℃, and the flow rate is 1.71m3·h-1”. Lines 149-150, Step 3 was presented. "The specific experimental conditions" (Page 4, Line 131) and "other experimental conditions" (Page 5, Line 140) are the same working conditions.

Point 6:  In Table 1:It is better to change “type” with “parameters”; The air volume is measured in m3/h? Replace “indoor temperature condition“ with “indoor temperature”. Page 6, Line 173, at the beginning of Section 3.2.1, replace “the influence” with “The influence”.  

Response 6: Thanks the respectable reviewer for your valuable comment. We applied your valuable comments to the manuscript to improve the quality of it. The air volume flow rate is measured in m3·h-1 .Thank you for your time and consideration.

Point 7: Page 7, Line 190, it is specified “It can be seen from the figure ….”. About what Figure is taking about? Page 8, Line 201, replace “air volume” with “air volume flow rate” considering that is talking about m3/h. Page 8, Line 203, replace “figure” with “Figure”. Page 8, Line 209, it is specified “It can be seen from the figure that under two different air volumes….”. Specify the number of the figure. Page 9, Line 228, replace “8” with “Figure 8”. The same remark at Line 269, pag.11. Page 10, Line 237, replace “;” with “.” at the end of the sentence. Furthermore, at Line 238 let a space between the sentences “than R22.At the optimal”. Page 11, Line 276, delete the point in “the optimal filling rate. Li [23] et al.”

Response 7: Thanks the respectable reviewer for your valuable comment. The Figure are Figures 6-7, and we applied your valuable comments to the manuscript to improve the quality of it.  

Point 8: Page 12, Line 293, is the “air inlet volume” or “air inlet volume flow rate”? Also, the sentence “need to be measured during the actual operation, which can” is incomplete. Furthermore, the sentence at the Lines 295-295 is too vague “The corresponding heat exchange of the system can be calculated quickly…”. Finally, at the Conclusions, the Authors should spread some words about the practical importance regarding the experimental results obtained in the literature. I strongly suggest that the Authors will carry out more studies to compare their results to that from other similar studies. 

Response 8: Thank you for your valuable question. In the unmodified paper, Page 12, Line 293, is the “air inlet volume flow rate”. In actual operation, we only need to measure the inlet air temperature, inlet air flow, inlet water temperature and inlet water flow, which are four physical quantities convenient for measurement. Then the corresponding heat transfer value of the system can be obtained by model (16). The error of the overall heat transfer prediction model of the system is about ± 12%, which also can basically meet the requirements for rapid prediction of the heat transfer of the system in practical applications. Calculation details are listed in 3.5 É›-NTU simplified model. We will build a steady-state model in the next study and compare it with others.

Reviewer 3 Report

This article discusses the results of a thermal analysis aimed at improving the refrigeration performance in data centers. This study is relevant, and can be accepted for publication provided that the following comments are addressed:

1- Define how the various indicators are calculated (EER, COP, PUE) and discuss their practical interpretation.

2- “It was founded” should be changed to “it was found”. This mistake is found in several places in the article. I would recommend to check thoroughly the grammar.

3- Given the raise of AI and other data-driven methods, the role of data centers will get more prominent. Therefore, finding more efficient ways of operating them will be essential. This is thoroughly discussed in this study, which should be added to the introduction: preprint arXiv:1905.00501, 2019

4- Mention the role of experimental uncertainties in the presented analysis.

5- Figures 1 and 10: Not necessary to have the legend if the same quantity is labeled on the y axis.

6- Figures 4 and 5: move the caption to the bottom of the figure.

7- Figures 8 and 9 are identical, but figure 8 should show the heat exchange according to the caption.

8- Can the authors make an assessment on the expected accuracy of the results using the NTU method? Would they extend this work in the future by performing CFD simulations?

9- State more clearly the type of data center under study, and how general the developed method is.

Author Response

Response to Reviewer 3 Comments

This article discusses the results of a thermal analysis aimed at improving the refrigeration performance in data centers. This study is relevant, and can be accepted for publication provided that the following comments are addressed:

Response: Dear reviewer, thank you for your valuable suggestions. We applied your kind comments to the manuscript to improve the quality of it. Thank you for your time and consideration.

Point 1: Define how the various indicators are calculated (EER, COP, PUE) and discuss their practical interpretation.

Response 1: Thanks the respectable reviewer for your valuable comment. The various indicators were defined and discussed in Nomenclature.

Point 2:“It was founded” should be changed to “it was found”. This mistake is found in several places in the article. I would recommend to check thoroughly the grammar.

Response 2: Thanks the respectable reviewer for your  valuable comment. The full text has been carefully checked and modified.

Point 3: Given the raise of AI and other data-driven methods, the role of data centers will get more prominent. Therefore, finding more efficient ways of operating them will be essential. This is thoroughly discussed in this study, which should be added to the introduction: preprint arXiv:1905.00501, 2019.

Response 3: Thanks the respectable reviewer for your valuable comment. We applied your valuable comments to the manuscript to improve the quality of it.

Point 4: Mention the role of experimental uncertainties in the presented analysis.

Response 4: Thanks the respectable reviewer for your valuable comment. The experimental uncertainties were presented and analyzed in Lines 153-158.

Point 5: Figures 1 and 10: Not necessary to have the legend if the same quantity is labeled on the y axis.

Response 5: Thanks the respectable reviewer for your valuable comment. We applied your valuable comments to the manuscript to improve the quality of it.

Point 6: Figures 4 and 5: move the caption to the bottom of the figure.

Response 6: Thanks the respectable reviewer for your valuable comment. We applied your valuable comments to the manuscript to improve the quality of it.

Point 7: Figures 8 and 9 are identical, but figure 8 should show the heat exchange according to the caption.

Response 7: Thank you for your time and consideration. This error has been corrected.

Point 8: Can the authors make an assessment on the expected accuracy of the results using the NTU method? Would they extend this work in the future by performing CFD simulations?

Response 8: Thanks the respectable reviewer for your valuable comment. The error of the overall heat transfer prediction model of the system is about ± 12%, which also can basically meet the requirements for rapid prediction of the heat transfer of the system in practical applications. Calculation details are listed in 3.5 ε-NTU simplified model. We will extend this work in the future by performing CFD simulations.

Point 9: State more clearly the type of data center under study, and how general the developed method is.

Response 9: Thank you for your valuable question. The system has been applied in China data center, include:

Dongjiang cloud nest data center, Changde xiananmen Telecom Data Center, Lugu Telecom and other various data centers.

The micro-channel backplane heat pipe system is mainly used in telecommunicati

Round 2

Reviewer 2 Report

The Authors have introduced some improvements in the quality of the paper and responded to all questions included in my previous revision. Therefore, I recommend the paper for publication on Applied Sciences.